# Chrono-Nutrition and Diet Quality in Adolescents with Delayed Sleep-Wake Phase Disorder

**DOI:** 10.3390/nu12020539

**Published:** 2020-02-19

**Authors:** Maxime Berendsen, Myrthe Boss, Marcel Smits, Gerda K. Pot

**Affiliations:** 1Division of Human Nutrition, Wageningen University & Research, 6700 AA Wageningen, The Netherlands; 2Hospital Gelderse Vallei, 6716 RP Gelderland, The Netherlands; 3Louis Bolk Institute, 3981 AJ Bunnik, The Netherlands; 4Department of Nutritional Sciences, King’s College London, London SE1 9NH, UK

**Keywords:** chrono-nutrition, meal timing, diet quality, delayed sleep-wake phase disorder, circadian rhythms, adolescents

## Abstract

**Background**: Delayed sleep-wake phase disorder (DSPD), characterized by delayed sleep-onset and problems with awakening in the morning, is mostly prevalent in adolescents. Several studies have suggested chrono-nutrition could present a possible modifiable risk factor for DSPD. Objective: To describe differences in chrono-nutrition and diet quality in adolescents with DSPD compared to age-related controls. **Methods***:* Chrono-nutrition and diet quality of 46 adolescents with DSPD, aged 13–20 years, and 43 controls were assessed via questionnaires. Diet quality included the Dutch Healthy Diet index (DHD-index) and Eating Choices Index (ECI). Results were analysed using logistic regression and Spearman’s partial correlation. **Results**: Compared with controls, DSPD patients consumed their first food of the day significantly later on weekdays (+32 ± 12 min, *p* = 0.010) and weekends (+25 ± 8 min, *p* = 0.005). They consumed their dinner more regularly (80.4% vs. 48.8%, *p* = 0.002) and consumed morning-snacks less frequently (3.0 ± 2.1 days vs. 4.2 ± 1.7 days, *p* = 0.006). No differences in clock times of breakfast, lunch, or dinner were found. Moreover, no significant differences in overall diet quality were observed. **Conclusion**: This descriptive study showed chrono-nutritional differences between adolescents with and without DPSD. Further studies are needed to explore features of chrono-nutrition as a possible treatment of DPSD.

## 1. Introduction

There is an increasing interest, besides *what* people eat, in *when* they eat [1]. The relationship between dietary intake and the circadian clock is called *chrono-nutrition*, which indicates the impact of the timing of eating on health [2,3,4]. Chrono-nutrition includes three components of time: (1) regularity, (2) frequency, and (3) clock time [5]. All three components could affect circadian rhythms as well as the other way around [6]. Light, hormones and timing of food intake are synchronising factors (also called ‘*zeitgebers*’) able to reset circadian clocks [1,7]. These factors could lead to circadian desynchronization when people are exposed to a factor at the wrong time for their internal clock, which might have a negative impact on their health [1,4,7,8].

To gain more insight in the impact of chrono-nutrition, patients with circadian rhythm sleep-wake disorders (CRSD) can be used as a model to study chrono-nutrition. Delayed Sleep-wake Phase Disorder (DSPD) frequently occurs in individuals who are otherwise healthy. DSPD especially affects adolescents and young adults (aged 10 to 24 years) with a prevalence between 7% and 16% [9,10]. DSPD is characterized by a delay in timing of sleep and often manifested as complaints of prolonged sleep onset latency and problems with awakening in the morning [11]. Moreover, DSPD is associated with lower school grades, smoking, alcohol usage and increased anxiety and depression scores [12]. A delayed endogenous 24 h melatonin rhythm is an important hallmark of the diagnosis of DSPD [13]. Currently, first-line treatments for DSPD consists of strengthening time cues, improving sleep hygiene, experiencing daylight in the morning, and avoidance of light at night [14,15]. When these measures prove to be insufficient, cognitive-behavioural therapy, well-timed and well-dosed melatonin and light therapy [16,17,18] can be used in an attempt to shift the body’s circadian rhythm and thereby improve daytime sleepiness, fatigue, and cognitive performance among individuals with DSPD [16,19].

As chrono-nutrition is a time-cue involved in circadian rhythmicity [7], time of meal consumption and diet quality might influence the development of CRSD such as DSPD. Consequently, these modifiable time cues might be used as possible treatment for CRSD as DSPD. However, research on this topic is scarce. Therefore, this study aims to investigate differences in chrono-nutrition and diet quality in adolescents aged 13 to 20 years with DSPD compared with age-related controls. It was hypothesised that DSPD patients eat less regularly, more often skip breakfast, and eat later at night [20,21,22]. Additionally, DSPD patients were expected to eat less healthily compared with controls [23,24,25,26].

## 2. Materials and Methods

### 2.1. Study Population

The study “Sleep, nUtrition, eating and delayed Sleep PhasE Disorder iN aDolesecents” (SUSPEND study), is an on-going observational study of adolescents visiting Hospital Gelderse Vallei for potential DSPD or other sleep disorders. This paper includes data that were collected between 2017 and 2019. The patient group consisted of adolescents aged 13 to 20 years who were diagnosed with DSPD. DSPD patients were recruited at the Sleep Centre of Hospital Gelderse Vallei (Ede, NL, USA) during their consultation with the neurologist-somnologist. The Sleep Centre of Hospital Gelderse Vallei is one of the few specialized and multi-disciplinary sleep centres in the Netherlands and covers a large part of Eastern Netherlands. Patients with other sleeping problems and patients who already used melatonin for a longer period of time (i.e., over 1 month) were excluded. Moreover, participants were excluded when important data was missing. 

The control group included adolescents aged 13 to 20 years without DSPD, who were not treated by a doctor and did not use melatonin. Controls were adolescents recruited via a high school, siblings or friends of adolescents with DSPD or patients without DSPD from the sleep centre of the Hospital Gelderse Vallei. 

The study was approved by the Beoordelings-Commissie Wetenschappelijk Onderzoek (BCWO) of Hospital Gelderse Vallei [27]. Participants provided written informed consent before completing the questionnaires. 

### 2.2. Diagnosis of DSPD

Before their hospital visit, patients completed online questionnaires about their sleeping habits and sleep quality. Moreover, adolescents performed a self-test at home to determine salivary time of dim-light melatonin onset (DLMO). Melatonin levels were measured in the evening for five consecutive hours. The lowest DLMO threshold for saliva is defined at 4 pg/mL and this melatonin concentration remains elevated at night [28]. An Actometer was used to confirm the time of sleep onset [29]. For controls we did not have DLMO data available.

### 2.3. Demographics

Information about the diagnosis of DSPD and general information about the patients such as sex, age, BMI, were received from the treating neurologist and the hospital system. Information about sex, age, education, height, weight, disorders, smoking habits, potential sleeping problems and melatonin use of the controls was obtained through a questionnaire. Height and weight were self-reported by the participants and BMI (in kg/m^2^) was subsequently calculated. BMI was also categorized into the following categories: underweight, normal weight, and overweight, based on age-specific national cut-offs [30]. Physical activity of the participants was assessed by two questions about the frequency and duration of moderate physical activity.

### 2.4. Assessment of Chrono-Nutrition

To assess chrono-nutrition, a 23-item questionnaire encompassing meal regularity, meal frequency and meal clock time was used. Questions were based on questionnaires and literature from other studies about the relationships between chrono-nutrition, metabolic syndrome, and the biological clock [31,32,33]. The questionnaire consisted of 8 closed questions on meal regularity, 7 closed questions on meal frequency, and 11 closed questions on clock time of meals. Meal regularity was defined as foods being consumed in varying amounts through the day and at different times from one day to the next [5].

### 2.5. Assessment of Diet Quality

A 48-item questionnaire was used to determine diet quality of the participants, based on the DHD-index score [34,35]. The DHD-index is a validated index, assessing adherence to the Dutch dietary guidelines of 2006 for a healthy diet according to the Health Council of the Netherlands [34,36]. The DHD-index consists of ten different components, namely physical activity; intakes of vegetables, fruit, fish, fibre, saturated fatty acids (SFA), and trans fatty acids (TFA); number of occasions of consumption of acidic drinks and foods (ADF), sodium, and alcohol. The DHD-index score for the number of consumption occasions of ADF was not included in this study, since variance for this component was low [34]. Therefore, the DHD-index score was only calculated for nine components. Each component of the DHD-index has a minimum score of 0 (no adherence to the Dutch dietary guidelines) and a maximum score of 10 (complete adherence to the Dutch dietary guidelines). This resulted in a total DHD-index score between 0 and 90 points [34]. 

The ECI score was calculated to measure the healthiness of the diet [37]. The ECI score is a validated measure that consists of four components: (1) consumption of breakfast, (2) consumption of two portions of fruit per day, (3) type of bread consumed, and (4) type of milk consumed. This score was originally developed for the UK population [37], but it is likely that the ECI score is applicable to most Western populations. Each component of the ECI score has a score from 1 to 5. After assigning the scores for the four different components, a total ECI score was calculated, ranging from 4 to 20. A score of 4 indicated an unhealthy diet and a score of 20 indicated the healthiest diet [37]. For non-bread consumption and no milk consumption, a neutral score of 3 was used. A score of 3 was also used for soy milk or other plant-based milk. 

### 2.6. Statistical Analyses

For the baseline characteristics of the adolescents, data were presented as means, ± standard deviations (SD), or as numbers (n) and percentages (%). For non-normal distributed data, the median and interquartile range (Q1–Q3) were used. First, overall mean differences in chrono-nutrition were tested between DSPD patients and controls. Differences in regularity of meal intake were tested using the chi-square test and differences in frequency and clock time of meals were tested using the Mann–Whitney U test. Additionally, logistic regression was used to study chrono-nutrition in DSPD patients compared with controls. Differences in diet quality between DSPD patients and controls were tested using independent samples t-test for total DHD-index score and ECI score and the Mann–Whitney U test for individual component scores. Moreover, logistic regression was used to study diet quality in DSPD patients compared with controls. Spearman’s partial correlation was used to study the correlation between chrono-nutrition (regularity, frequency, and clock time of meal intake) and diet quality (total DHD-index score and total ECI score) in DSPD patients and controls. Logistic regression and Spearman’s partial correlation were both adjusted for sex, age, education, BMI, and disorders. Data were missing for *n* 3 for height, weight, and BMI; *n* 1 for physical activity, DHD-index score and ECI score; *n* 9 for DLMO and *n* 12 for Actigraphy time point. In all analyses, *p* values < 0.05 were considered statistically significant. Data were analysed using the statistical program SPSS Statistics version 24 (IBM Corp., Armonk, NY, USA). 

## 3. Results

### 3.1. Characteristics of Study Population

This study included 46 DSPD patients and 43 controls (Table 1). The majority of patients and controls were girls. Mean (SD) DLMO of the DSPD patients was 23:15 (1:26) h. DLMO was not assessed in controls. DSPD patients had a higher median BMI of 21.3 kg/m^2^ (IQR 19.8–23.6) compared with controls with a median BMI of 19.7 kg/m^2^ (IQR 18.7–22.2). Overall, a higher proportion of DSPD patients had a lower educational level and a disorder (ADHD/ADD, autism or a combination of two disorders) compared with the control group. Both diet quality and physical activity were slightly lower in DSPD patients, but this was not statistically significant. The group of DSPD patients and the control group were similar in terms of age, height, smoking status, and alcohol consumption.

### 3.2. Chrono-Nutrition

Table 2 shows elements of chrono-nutrition, regularity, frequency, and clock time, in DSPD patients compared with controls. DSPD patients consumed their dinner significantly more regularly compared with controls (80.4% vs. 48.8%, *p* = 0.002). Moreover, DSPD patients consumed snacks in the morning significantly less frequently compared with controls (3.0 ± 2.1 days vs. 4.2 ± 1.7 days, *p* = 0.006). These significant differences remained statistically significant after adjusting for covariates. No differences were found in clock times when patients or controls consumed breakfast, lunch, or dinner. However, DSPD patients consumed their first food of the day significantly later on weekdays (+32 ± 12 min, *p* = 0.010) and weekends (+25 ± 8 min, *p* = 0.005) compared with controls. These significant differences did not remain significant after adjusting for covariates. Other relevant, but in this study not significant, differences were less regular consumption of breakfast and lunch, less frequent consumption of breakfast, and later consumption of the last food of the day on weekdays and weekends in DSPD patients compared with controls.

### 3.3. Diet Quality

Table 3 presents the total DHD-index score, total ECI score, and individual components of the DHD-index score and ECI score in DSPD patients compared with controls. A somewhat lower mean DHD-index score was found in DSPD patients (57.7 ± 10.8) compared with controls (60.5 ± 12.5), but this was not statistically significant (*p* = 0.263). DSPD patients had a lower DHD-index score for the components physical activity, intakes of vegetables, fibre, fish, SFA and TFA and a higher score for sodium and alcohol compared with controls. These differences were only statistically significant for fibre (*p* = 0.034) and remained significant after adjusting for covariates (*p* = 0.047). The higher DHD-index score for the component sodium in DSPD patients became statistically significant after adjusting for covariates (*p* = 0.033). Mean ECI score was slightly lower in DSPD patients (12.4 ± 2.4) compared with controls (12.8 ± 1.9), but this was not statistically significant (*p* = 0.370). DSPD patients had a lower ECI score for the components breakfast and bread and a slightly higher score for fruit and milk compared with controls. Nevertheless, these differences were not statistically significant. The slightly higher ECI score for the component milk in DSPD patients became statistically significant after adjusting for covariates (*p* = 0.034). Finally, a positive correlation was found between DHD-index score and ECI score in controls (*r* = 0.47, *p* = 0.003), but not in DSPD patients (*r* = 0.17, *p* = 0.307).

### 3.4. Chrono-Nutrition in Relation to Diet Quality

Correlations between diet quality and chrono-nutrition are presented in Table 4. The DHD-index score was positively correlated with breakfast regularity in controls (r = 0.43, *p* = 0.008), but not in DSPD patients (r = −0.21, *p* = 0.200). Additionally, the DHD-index score was negatively correlated with the frequency of dinner consumption (r = −0.40, *p* = 0.014) and clock time of lunch (r = −0.63, *p* = 0.005) in controls. No significant correlations were found between DHD-index score and chrono-nutrition in DSPD patients. ECI score was positively correlated with the frequency of breakfast consumption in both DSPD patients (r = 0.59, *p* < 0.001) and controls (r = 0.56, *p* < 0.001). Moreover, ECI score was positively correlated with the number of meals consumed daily in DSPD patients (r = 0.33, *p* = 0.045) and the frequency of morning snacks consumed in controls (r = 0.34, *p* = 0.041). No significant correlations were found between ECI score and regularity or clock times of breakfast, lunch, dinner, or snacks.

## 4. Discussion

This descriptive study in DPSD patients showed differences in chrono-nutrition between adolescents with and without DPSD. DSPD patients consumed their dinner significantly more regular and they consumed snacks in the morning significantly less frequently compared with controls. Moreover, DSPD patients consumed their first food of the day significantly later on week and weekend days compared with controls. DSPD patients had a somewhat lower overall diet quality but this was not statistically significant. In both patients and controls, we observed a positive correlation between ECI score and frequency of breakfast consumption, number of meals consumed daily (only in DSPD patients), and frequency of morning snacks consumed (only in controls). In controls, we observed a positive correlation between DHD-index score and breakfast regularity and a negative correlation between DHD-index score and the frequency of dinner consumption and clock time of lunch.

Our findings were partly in accordance with our hypothesis, based on previous studies on circadian preferences and eating behaviours, showing that evening persons are more likely to skip breakfast and have later meal start times [21,22,23]. DSPD patients are often evening persons, [38] but this was not specifically assessed in this study. Average breakfast consumption was less regular and less frequent in DSPD patients compared with controls, even though these differences were not significant. Moreover, DSPD patients consumed their first food of the day significantly later on week and weekend days compared with controls. DSPD patients often experience more problems with awakening in the morning, lack of appetite and less time in the morning, which could explain why they skipped or delayed their breakfast more often [11,33]. Additionally, a study by Nakade et al. [39] of 800 Japanese students aged 18–29 years showed that students who consumed breakfast at earlier and regular times were more often morning persons, compared to those who ate later or irregularly. The study also showed that regular consumption of breakfast seemed to be positively correlated to good sleep via the phase advance of the circadian clock. This indicates that breakfast possibly works as a ‘*zeitgeber*’ for the circadian clock [39].

DSPD patients consumed snacks in the morning significantly less frequently compared with controls, which could possibly be explained by the findings that they consumed breakfast less frequently and their first food of the day at a later time point. DSPD patients might have been less hungry in the middle of the morning and therefore less frequently consumed snacks in the morning. We could not come up with possible explanations as to why DSPD patients consumed their dinner significantly more regularly compared with controls. Patients did receive some advice about timing of meal intake during their consultation in the hospital, but the questionnaires they completed consisted of questions about their usual dietary intake of the past month. Moreover, dinner times are often mainly determined by the parents of the adolescents and by evening activities such as sports and music lessons [40]. Thus, DSPD patients and controls probably did not have much influence on their dinner regularity.

Our findings were not in line with our hypothesis that DSPD patients have a less healthy dietary pattern [24,25,26,27]. We found a somewhat lower overall diet quality in DSPD patients compared with controls, but this was not statistically significant. This suggests that these DSPD patients seemed to eat as healthily as controls. Nevertheless, DSPD patients had a higher BMI compared with controls, which indicates that it might also be important to consider when people eat instead of what they eat [7]. Follow up studies with a larger sample size should elaborate on this and other health-related outcomes could be taken into account in future studies, in order to gain more insight in the health of the participants [41]. We observed a positive correlation between DHD-index score and ECI score in controls, so both index scores seem to be a good measure for diet quality [34,37]. No correlation was observed between DHD-index score and ECI score in DSPD patients, therefore both measures for diet quality were used in this study. Only a few correlations between chrono-nutrition and diet quality were observed in this study, which might be due to our relatively small sample size. Therefore, further studies with a larger sample size should elaborate on this. Moreover, we would expect more differences between DSPD patients and controls if energy intake of the participants was also included [5,42]. Limited research has been published on chrono-nutrition in relation to dietary intakes and health outcomes in populations who do not perform shift work [43]. Nevertheless, previous studies in adolescents and (young) adults showed that regular breakfast consumption was associated with overall higher diet quality and skipping breakfast was associated with lower diet quality [42,44,45]. This is in line with the positive correlation found between DHD-index score and breakfast regularity in controls, suggesting a higher diet quality in controls who regularly consumed breakfast. The positive correlation between ECI and frequency of breakfast consumption in both DSPD patients and controls could be explained partly by the fact that breakfast consumption is one of the four components of ECI score [37]. This could also explain the positive correlation found between ECI score and the number of meals consumed daily in DSPD patients, since both ECI and the number of meals consumed will be higher when frequency of breakfast consumption is increased.

A higher proportion of DSPD patients in this study were girls, which is in line with the higher prevalence of sleep disorders among girls/women [46]. Girls and women have more problems with falling asleep and wake up more frequently during the night compared with boys and men [46]. A large study of university students examined sex differences in morningness and eveningness preferences and found significantly higher eveningness preference in men [47]. Similar results were found in other studies in young adults [48,49]. Nevertheless, eveningness does not always have to lead to a sleep disorder. DSPD patients had a higher BMI and consisted of a higher proportion of people with disorders compared with controls. A systemic review in adolescents demonstrated an increase in BMI in breakfast skippers compared with people who did consume breakfast [50], which corresponds with our findings. In addition, a study by Taheri et al. [51] showed that decreased sleep duration, which is often a problem in DSPD patients [52], was associated with increased BMI. Lower leptin levels and increased ghrelin levels were observed in participants with decreased sleep duration, which probably increases appetite and therefore results in a higher BMI [51]. Disorders such as autism spectrum disorders (ASD), attention deficit hyperactivity disorder (ADHD), and depression are more prevalent in patients with DSPD compared with individuals without DSPD. Sleeping problems are often observed at early onset of these psychiatric disorders [53,54]. This corresponds with the higher proportion of DSPD patients with disorders (ADHD/ADD, autism or a combination of two disorders) found in our study.

Our study has some limitations and strengths that need to be discussed. The questionnaire used in this study to assess diet quality is validated for anyone with a Dutch eating pattern, from 19 to 69 years old [35,55]. Therefore, this questionnaire might not have been suitable for adolescents between 13 and 20 years old. However, to our knowledge, no validated dietary questionnaires assessing diet quality are available for this age group. Since commencement of our study, a newer version of DHD-index, the DHD15-index, has come out which refers to the most recent Dutch dietary guidelines [34,55]. Furthermore, participants often completed the questionnaires in the presence of their parents, which could have resulted in more socially desirable answers from the participants. This was especially the case for DSPD patients but only for a small number of controls. Nevertheless, this is an issue with most dietary assessment methods. Only two questions were asked about physical activity in this study, which probably does not provide sufficient insight into the actual physical activity of the participants. Since physical activity may advance the circadian clock, future research should use objective measures of physical activity in order to correct for this covariate [56,57]. Another limitation of this study was that controls might not be representative for the general Dutch population between 13 and 20 years old. When comparing chrono-nutrition in controls with the Dutch population using the Dutch Nutrition and Food Survey (DNFCS) 2012–2014, [58] controls seemed to consume breakfast less frequently compared with Dutch adolescents aged 9–18 years old. However, there is a difference in age between the controls and DNFCS participants, since there was no dietary intake data available for the general Dutch population between 13 and 20 years old, and it is unknown whether participants in the DFNCS 2012–2014 had DSPD or other sleeping problems. For the controls in this study, there is a small chance of undiagnosed DSPD, however self-reported sleeping problems were limited and use of melatonin was an exclusion criterion. This is the first descriptive study on chrono-nutrition in patients with DPSD and controls included a limited sample size, which could have limited our power. Finally, findings may not be generalizable to DSPD patients of an older age or patients with other sleeping problems.

Our study also has several strengths. This was the first descriptive study on chrono-nutrition in DPSD patients, that investigated timing of meal intake and diet quality in DSPD patients. The diagnosis of DSPD was clinically determined using DLMO. Understanding how chrono-type influences or is influenced by chrono-nutrition and diet quality is important for the development of alternative treatment options of DSPD and other sleeping problems. Validated questionnaires were used in this study to collect data, which made results more reliable. DSPD is most prevalent in adolescents and young adults and this study was done in adolescents aged 13 to 20 years. Moreover, DSPD patients included in this study came from different locations in the Netherlands and had various ages, educational levels, and disorders. Thus, outcomes are representative for Dutch adolescent DSPD patients.

## 5. Conclusions

In conclusion, we observed chrono-nutritional differences between adolescents with and without DPSD pointing in the direction that their chrono-nutritional clock is also delayed. Nevertheless, this delay in the chrono-nutritional clock of DSPD patients was not evident for all meals. We did not observe differences in overall diet quality between DSPD patients and controls, which indicates that it might be more important when people eat instead of what they eat. Further studies are needed to explore features of chrono-nutrition as possible treatment of DPSD. Because breakfast might work as a ‘*zeitgeber*’ via the phase advance of the circadian clock, possible alternative treatment for adolescents with DSPD should focus on frequent and regular consumption of breakfast.

## Figures and Tables

**Table 1 nutrients-12-00539-t001:** Baseline characteristics of delayed sleep-wake phase disorder (DSPD) patients and controls of the Sleep, nUtrition, eating, and delayed Sleep PhasE Disorder iN aDolesecents (SUSPEND) study. Values are presented as mean ± SD, median (interquartile range) or as *n* (%).

Characteristics	DSPD Patients (*n* = 46)	Controls (*n* = 43)
Sex		
Girl	29 (63.0%)	24 (55.8%)
Boy	17 (37.0%)	19 (44.2%)
Age (years)	15.8 ± 1.5	15.5 ± 1.2
Height (cm) *	173 ± 9	175 ± 9
Weight (kg) *	62.5 (57.3–72.5)	62.0 (55.0–70.0)
BMI (kg/m^2^) ¹^,^ *	21.3 (19.8–23.6)	19.7 (18.7–22.2)
BMI categories, *n* (percentage) ^†^		
Overweight	2 (5%)	1 (2%)
Normal weight	31 (70%)	36 (88%)
Overweight	11 (25%)	4 (10%)
Educational level, *n* (percentage)		
Pre-vocational secondary education	19 (41.3%)	6 (14.0%)
Senior general secondary education	14 (30.4%)	21 (48.8%)
Pre-university education	8 (17.4%)	16 (37.2%)
Different type of education	5 (10.9%)	0 (0%)
Disorders, *n* (percentage)		
Yes, one disorder	18 (39.1%)	9 (21.0%)
Yes, a combination of disorders	15 (32.6%)	1 (2.3%)
No	13 (28.3%)	33 (76.7%)
Smoking status, *n* (percentage)		
Current smoker	3 (6.5%)	2 (4.7%)
Non-smoker	43 (93.5%)	41 (95.3%)
Consumes alcohol, *n* (percentage)		
Yes	7 (15.2%)	10 (23.3%)
No	39 (84.8%)	33 (76.7%)
Physical activity (day; min)	5.25 (3.30–5.30)	5.25 (4.15–5.30)
Diet quality		
DHD-index score ²^,^*	57.7 ± 10.8	60.5 ± 12.5
ECI score ³^,^*	12.4 ± 2.4	12.8 ± 1.9
DLMO *	23:15 ± 1:26	21:08 ± 1:13⁴
Actigraphy time point *	23:42 ± 1:10	NA

DLMO, Dim Light Melatonin Onset; DSPD, Delayed Sleep-Wake Phase Disorder; DHD-index, Dutch Healthy Diet Index; ECI, Eating Choices Index; NA, not assessed; * Height, weight, and BMI are missing for 2 patients; Weight and BMI are missing for 1 control; Physical activity is missing for 1 patient; DHD-index score and ECI score are missing for 1 patient; DLMO is missing for 9 patients; Actigraphy time point is missing for 12 patients.^1^ BMI was calculated using self-reported height and weight. ^2^ DHD-index score has a minimum of 0 and a maximum of 90. ^3^ ECI score has a minimum of 0 and a maximum of 20. ^4^ Reported by an earlier study in healthy adolescents [13]. ^†^ age specific BMI cut-offs were used.

**Table 2 nutrients-12-00539-t002:** Chrono-nutrition in DSPD patients compared with controls. Values are presented as mean ± SD or *n* (%).

Chrono-Nutrition	DSPD Patients (*n* = 46)	Controls (*n* = 43)	*p*-Value ¹	Adjusted *p*-Value ²
(1) Regularity				
Breakfast regularity			0.501	0.374
Regular	13 (28.3%)	15 (34.9%)		
Not regular	33 (71.7%)	28 (65.1%)		
Lunch regularity			0.171	0.471
Regular	19 (41.3%)	24 (55.8%)		
Not regular	27 (58.7%)	19 (44.2%)		
Dinner regularity			0.002 *	0.015 *
Regular	37 (80.4%)	21 (48.8%)		
Not regular	9 (19.6%)	22 (51.2%)		
Snacks morning regularity			0.064	0.164
Regular	9 (19.6%)	16 (37.2%)		
Not regular	37 (80.4%)	27 (62.8%)		
Snacks afternoon regularity			0.882	0.443
Regular	8 (17.4%)	8 (18.6%)		
Not regular	38 (82.6%)	35 (81.4%)		
Snacks evening regularity			0.116	0.22
Regular	14 (30.4%)	7 (16.3%)		
Not regular	32 (69.6%)	36 (83.7%)		
General regularity			0.428	0.629
Regular	6 (13.0%)	3 (7.0%)		
Most days regular	28 (61.0%)	24 (55.8%)		
Some days regular	6 (13.0%)	11 (25.6%)		
Not regular	6 (13.0%)	5 (11.6%)		
(2) Frequency				
Breakfast days	5.3 ± 2.4	6.3 ± 1.3	0.115	0.917
Lunch days	5.7 ± 1.5	6.2 ± 1.0	0.177	0.976
Dinner days	6.9 ± 0.3	7.0 ± 0.2	0.169	0.135
Snacks morning days	3.0 ± 2.1	4.2 ± 1.7	0.006 *	0.033 *
Snacks afternoon days	4.5 ± 2.3	4.5 ± 2.1	0.997	0.148
Snacks evening days	4.7 ± 2.3	4.3 ± 2.1	0.325	0.259
Number of meals per day	2.5 ± 1.4	2.7 ± 1.2	0.945	0.311
(3) Clock time				
Breakfast time	7:48 ± 1:00	7:32 ± 0:28	0.387	1
Lunch time	12:37 ± 0:34	12:50 ± 0:26	0.147	0.47
Dinner time	17:58 ± 0.41	18:25 ± 0:58	0.141	0.857
Snacks morning time	10:10 ± 1:11	10:19 ± 1:06	0.548	1
Snacks afternoon time	15:41 ± 0:31	14:56 ± 1:38	0.328	ND
Snacks evening time	20:15 ± 0:46	20:21 ± 0:45	0.743	ND
First food week days	8:18 ± 1:05	7:45 ± 0:51	0.010 *	0.904
First food weekend days	9:37 ± 0:36	9:12 ± 0:47	0.005 *	0.139
Last food week days	21:09 ± 1:08	20:41 ± 1:15	0.082	0.944
Last food weekend days	21:52 ± 1:03	21:32 ± 1:11	0.144	0.695

DSPD, Delayed Sleep-Wake Phase Disorder; ND, Not determined.; ¹ *p*-values were determined using Chi-Square test for regularity and Mann-Whitney U test for frequency and clock time. ² *p*-values were determined using logistic regression adjusting for sex, age, education, BMI and disorders. * Statistical significance at *p* < 0.05.

**Table 3 nutrients-12-00539-t003:** Total DHD-index score, total Eating Choices Index (ECI) score and individual components of the DHD-index score and ECI score in DSPD patients compared with controls. Values are presented as mean ± SD.

Variable	DSPD Patients (*n* = 45)	Controls (*n* = 43)	*p*-Value ¹	Adjusted *p*-Value ²
Total DHD-index score ³^,^†	57.7 ± 10.8	60.5 ± 12.5	0.263	0.550
Physical activity	6.7 ± 3.3	7.6 ± 3.1	0.173	0.165
Vegetable	4.2 ± 2.5	4.7 ± 2.6	0.408	0.747
Fruit	6.6 ± 3.4	6.6 ± 2.9	0.879	0.467
Fibre	6.5 ± 2.5	7.6 ± 1.8	0.034*	0.047*
Fish	3.4 ± 3.3	3.7 ± 3.4	0.787	0.149
SFA	6.9 ± 4.2	7.1 ± 3.7	0.790	0.121
TFA	8.2 ± 3.9	8.8 ± 3.2	0.419	0.275
Sodium	6.0 ± 3.0	5.5 ± 2.8	0.300	0.033*
Alcohol	9.1 ± 2.9	8.8 ± 3.2	0.673	0.504
Total ECI score ⁴^,^†	12.4 ± 2.4	12.8 ± 1.9	0.370	0.371
Breakfast component	3.9 ± 1.3	4.4 ± 0.9	0.162	0.690
Fruit component	3.0 ± 1.0	2.9 ± 0.8	0.496	0.988
Milk component	2.9 ± 0.4	2.8 ± 0.6	0.503	0.034*
Bread component	2.5 ± 1.2	2.8 ± 1.0	0.216	0.862

DSPD, Delayed Sleep-Wake Phase Disorder; DHD-index, Dutch Healthy Diet Index; ECI, Eating Choices Index; SFA, Saturated fatty acids; TFA, Trans fatty acids. Only total DHD-index score and total ECI score were normally distributed. ^1^
*p*-values were determined using independent samples t-test for total DHD-index score and ECI score and Mann-Whitney U test for individual component scores. ^2^
*p*-values were determined using logistic regression adjusting for sex, age, education, BMI, and disorders. ^3^ Total DHD-index score has a minimum of 0 and a maximum of 90; Individual components from the DHD-index have a minimum score of 0 and a maximum of 10. ^4^ Total ECI score has a minimum of 0 and a maximum of 20; Individual components from the ECI score have a minimum of 0 and a maximum of 5. ^†^ A positive correlation was found between DHD-index score and ECI score in controls (r = 0.47, *p* = 0.003), but not in DSPD patients (r = 0.17, *p* = 0.307). Spearman’s partial correlation was used adjusted for sex, age, education, BMI and disorders. * Statistical significance at *p* < 0.05.

**Table 4 nutrients-12-00539-t004:** Correlation between chrono-nutrition and diet quality in DSPD patients compared with controls. Spearman’s partial correlation adjusted for sex, age, education, BMI and disorders).

Chrono-Nutrition	DSPD Patients (*n* = 43) ¹				Controls (*n* = 42) ¹			
	DHD-Index Score		ECI Score		DHD-Index Score		ECI Score	
	r	*p*	r	*p*	r	*p*	r	*p*
**(1) Regularity**								
Breakfast regularity	−0.21	0.2	−0.01	0.97	0.43	0.008 *	0.18	0.297
Lunch regularity	0.12	0.484	−0.26	0.119	0.05	0.763	0.01	0.952
Dinner regularity	−0.06	0.717	−0.07	0.696	−0.27	0.101	−0.04	0.802
Snacks morning regularity	−0.21	0.197	−0.31	0.062	0.02	0.893	−0.11	0.529
Snacks afternoon regularity	0.08	0.651	0.07	0.669	0.16	0.339	0.17	0.328
Snacks evening regularity	0.08	0.634	0.1	0.551	0.02	0.921	0.1	0.576
General regularity	0.06	0.738	−0.15	0.367	0.09	0.582	−0.08	0.629
(2) Frequency								
Breakfast days	−0.06	0.706	0.59	0.000 *	0.2	0.241	0.56	0.000 *
Lunch days	−0.17	0.3	0.08	0.631	0.04	0.802	0.18	0.295
Dinner days	−0.17	0.3	−0.21	0.2	−0.40	0.014 *	−0.14	0.401
Snacks morning days	0.04	0.836	0.08	0.625	0.05	0.786	0.34	0.041 *
Snacks afternoon days	−0.07	0.679	0.04	0.798	0.14	0.414	0.24	0.156
Snacks evening days	−0.29	0.074	0.03	0.857	−0.25	0.141	0.04	0.804
Number of meals per day	−0.14	0.388	0.33	0.045 *	0.18	0.295	0.16	0.344
(3) Clock time								
Breakfast time	−0.36	0.388	0.27	0.59	−0.28	0.465	−0.07	0.854
Lunch time	0.36	0.221	0.1	0.748	−0.63	0.005 *	−0.13	0.617
Dinner time	0.07	0.742	0.1	0.632	−0.11	0.7	0.03	0.915
Snacks morning time	ND		ND		−0.17	0.641	0.25	0.48
Snacks afternoon time	ND		ND		0.61	0.581	−0.68	0.522
Snacks evening time	0.08	0.859	−0.04	0.922	ND		ND	
First food week days	−0.30	0.071	−0.02	0.915	−0.21	0.208	−0.28	0.093
First food weekend days	0.23	0.165	−0.18	0.284	−0.24	0.15	−0.10	0.543
Last food week days	−0.01	0.977	−0.11	0.525	−0.02	0.924	−0.05	0.763
Last food weekend days	−0.01	0.952	−0.08	0.642	−0.07	0.676	−0.12	0.488

DSPD, Delayed Sleep-Wake Phase Disorder; DHD-index, Dutch Healthy Diet Index; ECI, Eating Choices Index; ND, Not determined. ^1^ BMI is missing for 2 patients and 1 control; DHD-index score and ECI score are missing for 1 patient. * Correlation is statistically significant at *p* < 0.05.

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
