# Peer review of "Chrono-Nutrition and Diet Quality in Adolescents with Delayed Sleep-Wake Phase Disorder"

_nutrients, 2020, doi:10.3390/nu12020539_

Round 1

Reviewer 1 Report

Dear Editor,

I carefully read the paper by Berendsen et al. which is overall well-written and balanced in its parts. The current study yield interesting results, even if I think that there are some methodological flaws which weaken the conclusions.

Some comments for the authors:

how was the sample size assessed? Table 1 - How was the existence of differences between the two groups assessed? The study groups seem different to me at least in terms of schooling, for example. Which was the statistical test used? I think that the authors might update and correct the section of statistical methods accordingly to my suggestion. Line 304-305: In my opinion, this is a great limitation, which in fact invalidates the results obtained from the study. Even if no validated dietary questionnaires assessing diet quality are available for children with age <19 years, authors cannot use a questionnaire which has been validated for another age group!

Author Response

We kindly thank the referees and associate editor for their evaluation of our manuscript. The comments were valuable to further the manuscript. Please find our point by point reply below. We have highlighted the changes made in the manuscript with track changes. 

Point by point Reply:

How was the sample size assessed?

Reply: As this is a descriptive observational study with a new research question that has not been investigated elsewhere, we could not conduct a power analysis.

Table 1 - How was the existence of differences between the two groups assessed? The study groups seem different to me at least in terms of schooling, for example. Which was the statistical test used? I think that the authors might update and correct the section of statistical methods accordingly to my suggestion.

Reply: As this was the first observational study on chrono-nutritional differences between DSPD patients and controls we only used descriptive statistics to assess differences between these two groups. For this we used descriptive statistics, including chi-square for categorical variables and Mann Whitney U for numerical variables as described in the methods section. Furthermore, differences in diet quality (the primary outcome of this study) between DSPD patients and controls were tested using independent samples t-test as well as logistic regression models, adjusting for sex, age, education, BMI as disorders as we had recognised that these were potentially different between DSPD cases and controls.

Line 304-305: In my opinion, this is a great limitation, which in fact invalidates the results obtained from the study. Even if no validated dietary questionnaires assessing diet quality are available for children with age <19 years, authors cannot use a questionnaire which has been validated for another age group! 

Reply: We recognize in the Discussion that the fact the dietary intake questionnaires were not validated for this specific age group as a limitation of the study. However, this does mean that this method cannot be used in this study population. Moreover, currently in the Netherlands there are no other brief questionnaires available to assess diet quality in this age group. This study provides the first insights into chrono-nutritional differences in adolescents with a sleep-wake phase disorder, and we acknowledge this potential limitation, we think this paper provides novel insights and is therefore still relevant.

Reviewer 2 Report

This is an excellent report of a new study population with a high degree of societal relevance. The authors used sufficient and necessary physiological and subjective end point measures to address their hypothesis. Results are nicely presented and the scope of eating habits, diets, and general anthropometric analyses is extremely comprehensive. Well done!!

Author Response

Thank you for your positive feedback.

Reviewer 3 Report

This is an interesting study concerning on how chrono-nutrition relates to sleep disorders, namely delayed sleep-wake phase disorders (DSPD) in adolescents. More information on some of the methodology will improve the quality of the study. My specific comments are below

Specific comments

The authors discuss the possibility of a lack of power in their associations. Did they perform some calculation of sample size that could back up this justification?

Materials and methods

Could the authors add some information regarding the coverage of the hospital where participants were recruited?

Regarding the control group: some were patients from the sleep centre without DSPD. Which diseases were they been treated for? Could that influence the studied associations? They might have had a non-diagnose DSPD?

It is only in the results that it is explained that DLMO was not assessed in the control group (lines137-138). That should be clarified in the methods section 2.2

Was BMI calculated using some reference values (ex. from WHO), taking into consideration the age and sex of the adolescents? Moreover, the inclusion of categories (ex. normal weight, overweight..) may be more informative than only the BMI median (ex. table 1)

Lines 66: What is considered a long period of use of melatonin?

Section 2.4: Some explanation regarding the questionnaire used to evaluate the chrono-nutrition is missing. Is the questionnaire available to consult? Could the authors add some examples, explain the structure of the questionnaire (ex. open questions, close questions, Likert scale..?), etc

Lines 95-97: The information seems to be repeated. The authors could start explaining the DHD-index and only afterwards the ECI.

Table 2- What is considered by the authors meal regularity? For ex. having that meal every day?

Discussion, Line 267: Do the authors have some justification for the lack of association between DHD-index and the ECI score only in the DSPD patients?

Author Response

We kindly thank the referees and associate editor for their evaluation of our manuscript. The comments were valuable to further the manuscript. Please find our point by point reply below. We have highlighted the changes made in the manuscript with track changes. 

Point by point Reply:

This is an interesting study concerning on how chrono-nutrition relates to sleep disorders, namely delayed sleep-wake phase disorders (DSPD) in adolescents. More information on some of the methodology will improve the quality of the study. My specific comments are below

Specific comments

The authors discuss the possibility of a lack of power in their associations. Did they perform some calculation of sample size that could back up this justification?

Reply: As this is a descriptive observational study with a new research question that has not been investigated elsewhere, we could not conduct a power analysis.

Materials and methods

Could the authors add some information regarding the coverage of the hospital where participants were recruited?

Reply: We have added more information regarding the coverage of the hospital at section 2.1. The Sleep centre of Hospital Gelderse Vallei is one of the few specialised sleep centres in the Netherlands and includes patients from the Eastern part of the Netherlands.

Regarding the control group: some were patients from the sleep centre without DSPD. Which diseases were they been treated for? Could that influence the studied associations? They might have had a non-diagnose DSPD?

Reply: Some controls included in this study visited the hospital for sleeping problems whilst others were recruited through other channels, like siblings of patients or via schools. DLMO and an Actometer were used to determine the time of sleep onset in these controls visiting the hospital, to determine if the controls had DSPD. When it was established that these controls had no DSPD, they were asked to participate in the study as a control. Nevertheless, there is a chance that other controls could have a non-diagnosed DSPD. We have added this to the Discussion at lines 326-8.

It is only in the results that it is explained that DLMO was not assessed in the control group (lines137-138). That should be clarified in the methods section 2.2

Reply: We have added this information to the Methods section 2.2.

Was BMI calculated using some reference values (ex. from WHO), taking into consideration the age and sex of the adolescents? Moreover, the inclusion of categories (ex. normal weight, overweight..) may be more informative than only the BMI median (ex. table 1)

Reply: Thank you for the suggestion of including BMI categories (i.e. normal weight, overweight etc.). We have added the Dutch age specific BMI cut-offs for underweight, normal weight and overweight. We have added this to Table 1 and to the Methods at section 2.3.

Lines 66: What is considered a long period of use of melatonin?

Reply: We considered over 1 month as a long period and this information has been added to section 2.1

Section 2.4: Some explanation regarding the questionnaire used to evaluate the chrono-nutrition is missing. Is the questionnaire available to consult? Could the authors add some examples, explain the structure of the questionnaire (ex. open questions, close questions, Likert scale..?), etc

Reply: We have added information on the questionnaire to evaluate chrono-nutrition to section 2.4.

The chrono-nutrition questionnaire included 8 closed questions on meal frequency, 7 closed questions on meal regularity and 11 closed questions on clock time of meals (NB some have a degree of dependency so if one question is answered negative a subsequent question is skipped, therefore the total adds up to more than 23).  

Lines 95-97: The information seems to be repeated. The authors could start explaining the DHD-index and only afterwards the ECI.

Reply: We have adjusted this.

Table 2- What is considered by the authors meal regularity? For ex. having that meal every day?

Reply: Meal regularity was defined as a low day to day variability based on a previous paper by Pot et al (IJO 2014) . We have added this information to section 2.4 of the Methods.

Discussion, Line 267: Do the authors have some justification for the lack of association between DHD-index and the ECI score only in the DSPD patients?

Reply: Thank you for this question. We currently have no justification for the lack of association between the DHD-index and ECI score in DSPD patients. This is the first observational study in which these scores are used at the same time thus more studies are needed, perhaps also in different populations, to look into this further.

Round 2

Reviewer 1 Report

Dear Editor,

unfortunately the answers of the authors to my concerns have been inadequate. In my opinion, the current study has several methodological flaws which invalidate the findings. For this reason, I do not recommend the acceptation of the paper in the Journal.

Reviewer 3 Report

The authors have addressed all my questions